# New Amorphous Hydrogels with Proliferative Properties as Potential Tools in Wound Healing

**DOI:** 10.3390/gels8100604

**Published:** 2022-09-21

**Authors:** Petruta Preda, Ana-Maria Enciu, Bianca Adiaconita, Iuliana Mihalache, Gabriel Craciun, Adina Boldeiu, Ludmila Aricov, Cosmin Romanitan, Diana Stan, Catalin Marculescu, Cristiana Tanase, Marioara Avram

**Affiliations:** 1National Institute for Research and Development in Microtechnologies—IMT Bucharest, 126A Erou Iancu Nicolae, 077190 Bucharest, Romania; 2Biochemistry-Proteomics Department, Victor Babes National Institute of Pathology, 99-101 Splaiul Inde Pendenţei, 050096 Bucharest, Romania; 3Cell Biology and Histology Department, Carol Davila University of Medicine and Pharmacy, 8 Eroii Sanitari, 050474 Bucharest, Romania; 4“Ilie Murgulescu” Institute of Physical Chemistry, Romanian Academy, Spl. Independentei 202, 060021 Bucharest, Romania; 5DDS Diagnostic SRL, 7 Vulcan County, 031427 Bucharest, Romania; 6Doctoral School of Medicine, Titu Maiorescu University, Dâmbovnicului St., 040441 Bucharest, Romania; 7Faculty of Medicine, Titu Maiorescu University, Gheorghe Petrascu St., 031593 Bucharest, Romania

**Keywords:** polymeric biocomposites, bioactive compounds, green synthesis, cell proliferation, antibacterial activity

## Abstract

The study and discovery of bioactive compounds and new formulations as potential tools for promoting the repair of dermoepidermal tissue in wound healing is of continuing interest. We have developed a new formulation of amorphous hydrogel based on sodium alginate (NaAlg); type I collagen, isolated by the authors from silver carp tails (COL); glycerol (Gli); Aloe vera gel powder (AV); and silver nanoparticles obtained by green synthesis with aqueous *Cinnamomum verum* extract (AgNPs@CIN) and vitamin C, respectively. The gel texture of the amorphous hydrogels was achieved by the addition of Aloe vera, demonstrated by a rheological analysis. The evaluations of the cytotoxicity and cell proliferation capacity of the experimental amorphous hydrogels were performed against human foreskin fibroblast Hs27 cells (CRL-1634-ATCC). The developed gel formulations did not show a cytotoxic effect. The hydrogel variant containing AgNPs@CIN in a concentration of 8 µg Ag/gel formulation and hydrogel variant with vitamin C had proliferative activity. In addition, the antibacterial activity of the hydrogels was evaluated against *S. aureus* ATCC 6538, *Ps. aeruginosa* ATCC 27853, and *E. coli* ATCC 25922. The results demonstrated that the gel variant based on AgNPs@CIN in a concentration of 95 µg Ag/gel formulation and the hydrogel based on vitamin C show antibacterial activity. Therefore, the developed hydrogels with AgNPs@CIN and vitamin C could be promising alternatives in wound healing.

## 1. Introduction

Healing skin wounds is a dynamic and complex process that comprises five main physiological phases: homeostasis (blood clotting), inflammation, proliferation, cell migration, and tissue remodeling (maturation) [1,2]. Depending on the affected skin layer, the wounds/lesions on the skin are classified as superficial wounds (only the epidermal surface is affected) and deep wounds (epidermal, dermal, and hypodermic tissues are affected), and depending on the healing process, these can be acute wounds or chronic wounds. Acute wounds are usually lesions that heal completely within in 8–12 weeks and with minimal scarring. These types of wounds consist of mechanical skin lesions (cracks), penetrating lesions (cuts and gunshots), and surgical incisions. Another category of acute injuries is slight burns [2]. Chronic wounds are lesions that heal gradually over 12 weeks and often recur. Both acute and chronic wounds can present the risk of microbial contamination, with a higher risk in the case of chronic wounds due to the healing time. The healing of chronic wounds can be hampered by microbial contamination, by patient comorbidities (diabetes, obesity, cancer, etc.), drug administration with immunosuppressants, anticoagulants and corticosteroids, or repeated trauma to the affected area. The most common chronic wounds are foot ulcers (of a venous, ischemic, or traumatic origin), diabetic foot ulcers, pressure ulcers (ulcers), wounds caused by tumors, wounds associated with immunological diseases, deep burns, and skin grafts [1,2].

The choice of the type of treatment is made according to the type of wound, the surface, the depth, the location, the amount of exudate released, and the infection process. Liquid and semi-solid formulations, such as solutions, creams, and ointments, are widely used for wound healing, disinfection, cleaning, and debridement. Their structures limit their application in certain clinical phases of lesions, so they have been replaced with advanced products that include wound dressings and permanent skin replacements/substitutes [3].

Globally, modern or advanced polymeric dressings such as films, hydrogels, gels, foam dressings, porous dressings (sponges), and innovative nano/microfiber dressings (nano/microfiber membrane) were developed in order to create the optimal conditions for the healing process, whose properties can be improved by mixing with bioactive substances (such as natural compounds, vitamins, drugs, cell growth factors, metal nanoparticles, metal oxides, etc.). Of these, hydrogel dressings, made of hydrophilic, inflatable, insoluble, and biodegradable polymeric materials, can be used as a matrix for the controlled release of bioactive compounds in order to increase the effectiveness of therapy. The most common physical forms of hydrogels are sheets, porous matrices, amorphous hydrogels (gel), and hydrogel-impregnated gauze [4,5].

Amorphous hydrogels do not have a fixed shape; these hydrogels can be applied uniformly over the wound due to their physical properties [6]. They can be used to deliver bioactive compounds, for example, silver nanoparticles (AgNPs), known to be effective against various pathogens. Furthermore, the green synthesis of AgNPs—based on plant extracts rich in flavonoids, phenolic compounds, amino acids, vitamins, polysaccharides, proteins, and enzymes, which act as both reducing and stabilizing agents [7,8,9]—is simple, environmentally friendly, and inexpensive. Moreover, the nature of the biomolecules on the surface of silver nanoparticles can improve the bioactivity and biocompatibility of silver nanoparticles [7].

Current research has shown that various products based on silver nanoparticles (AgNPs), such as dressings, creams, gels, etc., are safe and approved for human skin [8,10]. AgNPs also have healing properties, facilitating the proliferative properties of dermal cells, which leads to faster wound healing. This prevents scarring and the healed skin has a smooth appearance [11]. Martínez-Higuera et al. (2021) studied the regenerative effect of Carbopol-based hydrogel with AgNPs synthesized using *Mimosa tenuiflora* extract on second-degree burns [12]. The gel proved to be promising in treating these wounds as a result of its bactericidal and anti-inflammatory properties. In addition, Carbopol-based hydrogel and AgNPs synthesized with aqueous extract of *Ammania baccifera* have been reported by Kiran et al. (2016) for the treatment and healing of burns [13]. Another formulation of amorphous hydrogel based on sodium alginate, gelatine and AgNPs (obtained by the green method) with antimicrobial and regenerative properties was reported by Diniz et al. (2020) [14].

The aqueous bark extract of *Cinnamomum verum* is one possible source of reducing and stabilizing agents for green NPs synthesis due to the constituents from the extract such as flavonoids and phenolic compounds [15]. The bark of *Cinnamomum verum* contains numerous bioactive constituents such as cinnamaldehyde, β-caryophyllene, 1–8 cineole, cinnamyl acetate, eugenol, linalool and terpenoids–phenylpropanoids, monoterpenes, sesquiterpenes, etc., which provide anti-inflammatory, antibacterial, antioxidant, antifungal, antitumor, analgesic, gastroprotective, anticancer, and wound-healing properties [16,17].

Aloe vera (*Aloe barbadensis*) is a very attractive plant in tissue repair because the bioactive compounds identified in the gel promote the cells’ attachment, migration, proliferation, and development. Aloe vera gel is used to treat wounds and burns, and can stimulate the proliferation of fibroblasts, the synthesis of collagen and hyaluronic acid, and promote angiogenesis. The phytochemical compounds from aloe vera gel involved in wound healing are mainly mucopolysaccharides (acemannan and glucomannan), followed by gibberellin (a growth hormone), amino acids, enzymes, sterols, minerals, and anthraquinone [6,18,19,20].

Sodium alginate (NaAlg) is an inexpensive natural polymer and has the ability to form a hydrogel. It is frequently used in wound dressing, drug delivery, and tissue engineering due to its biocompatibility, biodegradability, and nontoxicity [21]. Novel alginate-based wound-healing dressings such as hydrogel, sponges, and electrospinning mats have been mixed with synthetic or natural polymers (such as hyaluronic acid, polyvinyl alcohol, gelatine, collagen, chitosan, etc.) and bioactive compounds [22].

Collagen is a protein, a biocompatible natural polymer, with multiple applications in the medical and pharmaceutical fields. Type I collagen (about 90% of all body collagen in vertebrates) is found mainly in the skin, tendons, ligaments, and bones. This type of collagen is used as a biomaterial in tissue repair, cosmetics, and the pharmaceutical industry [23], and can be isolated from the skin, tendons, ligaments, and bones of cattle, rats, pigs, sheep, poultry, etc. [24]. Recently, fish collagen has attracted attention as a biomaterial due to its easy and cheap extraction from fish by-products (skin, fins, and scales), high biocompatibility, and low risk of transmitting genetic diseases [23].

Glycerol (propane-1,2,3-triol; trihydroxy propane; C_3_H_8_O_3_) is a trihydroxy alcohol that is frequently used in topical dermatological preparations. Glycerol has a keratolytic effect, anti-irritant activity, the ability to provide elasticity to the skin, moisturizing properties, and can accelerate the process of wound healing [25].

Vitamin C (ascorbic acid) is a natural antioxidant [26] that is useful in chronic wounds healing where there is a high level of reactive oxygen species (ROS) that delays healing. Vitamin C promotes the migration and proliferation of epidermal cells and the synthesis of collagen by fibroblasts [27].

In this paper, we aim to develop amorphous hydrogel formulations based on sodium alginate (NaAlg); type I collagen, which has been isolated in-house from silver carp tails (COL) (for more information see Preda et al., 2021) [28]; glycerol (Gli); Aloe vera powder (AV); and silver nanoparticles synthesized with aqueous *Cinnamomum verum* extract (AgNPs@CIN) and vitamin C, respectively, as potential tools for wound healing. We investigated the in vitro cell proliferation and antibacterial activity of various gel formulations. To our knowledge, no data have been reported thus far showing the proliferative or antibacterial activity of an amorphous hydrogel based on AgNPs obtained by green synthesis with aqueous *Cinnamomum verum* extract. Moreover, amorphous hydrogel formulations based on AgNPs@CIN and vitamin C have also not been reported thus far.

## 2. Results and Discussion

### 2.1. The Physical-Chemical Properties of Silver Nanoparticles

The synthetic process of AgNPs can be examined by UV–vis spectrometry due to the specific surface plasmon resonance (SPR) of nanoparticles in the visible range (400–500 nm), where the peak presence in the stated field confirms the presence of the AgNPs’ formation [17].

Therefore, the formation of AgNPs obtained with aqueous Cinnamomum verum extract has been confirmed by UV–vis spectroscopy. The SPR specific absorption spectrum for AgNPs@CIN was recorded at 413 nm (Figure 1a).

The SEM morphology images from Figure 1b show that the AgNPs@CIN have a predominantly spherical shape.

From the XRD analysis of the AgNPs@CIN, multiple X-ray diffraction peaks can be observed (Figure 1c). These are located at 38.05°, 44.46°, 64.36°, 77.38°, and 81.33° and correspond to (111), (200), (220), (311), and (222) reflections of silver with a face-centered cubic crystal structure (fcc) according to the ICDD database (card no. 001-1184) and the literature [18]. In addition, the intense and narrow reflection from 52.07° and the broad feature from 55.37° are given by the SiO2 substrate, which served as support for the XRD investigations. The crystal quality of the silver was assessed via the mean crystallite size, determined as 7.3 nm via the Scherrer equation and as 9.2 nm from the Williamson–Hall size–strain plot (blue line in Figure 1d). At the same time, a small lattice strain, ε, was found around 0.08%. In this method, the size broadening depends on 1⁄(cos θ), while the strain broadening depends on tan θ [29]. The shape factor of the AgNPs was taken, k = 0.9, assuming a spherical shape, as it is revealed from SEM micrograph.

Dynamic and electrophoretic light-scattering (DLS/ELS) measurements were taken in order to obtain information about the nanoparticles’ dispersion, their hydrodynamic diameters and dispersity state (in terms of polydispersity index—PI), and their colloidal stability, in terms of Zeta potential (ζ). According to the distribution graph presented in Figure 2, the mean hydrodynamic diameter of the obtained AgNPs@CIN is 40.05 nm, with a PI of 0.298, thus indicating a good monodispersity. Furthermore, the Zeta potential revealed negatively charged silver nanoparticles (ζ = −15.07 mV) (coated with electronegative biomolecule) [15] and a good stability. PI is a parameter that provides information about the characteristic particles; if the PI is less than 0.300, then the particles are monodispersed, and if it is higher than that the particles are polydispersed, with a tendency to agglomerate. A Zeta potential greater than +25 mV or less than −25 mV shows the high degree of stability of the nanoparticles [30].

Therefore, the UV–vis, SEM, XRD, and DLS analyses confirmed the acquirement of AgNPs by the green synthesis method described above.

### 2.2. Antibacterial Activity of Hydrogel Formulations

All the obtained experimental samples are transparent, but the AgNPs@CIN-based hydrogel acquired a slightly brown color attributed to the silver nanoparticles. The pH was also evaluated for the obtained hydrogel by the classical method, that is, with pH indicator paper (Carl Roth, Germany). The change in pH was observed as the compounds were added to the polymer mixture. Thus, NaAlg (2%) has a weakly acidic pH (pH = 6), and the addition of type I fish collagen changes the pH from weakly acidic to a neutral pH. This pH value is maintained after the addition of glycerol. When adding the AV solution, the pH changes from neutral to a moderately acidic pH (pH = 5), whereas the addition of AgNPs@CIN did not significantly change the pH values—they remained in a pH range of about 5–5.5. The addition of vitamin C to the BASE version determined the more acidic character of the gel; therefore, the hydrogel variant based on vitamin C showed a pH of 4–4.5.

In general, wound care products that provide a slightly acidic environment in the wound bed can improve wound healing. A slightly acidic pH has been shown to help heal wounds by controlling wound infections, altering the destructive activity of enzymes, releasing oxygen, and improving epithelialization and angiogenesis [31].

The antibacterial activity of the hydrogels was evaluated against the Gram-positive bacterial strain (*Staphylococcus aureus* ATCC 6538) and against the Gram-negative bacterial strains (*Pseudomonas aeruginosa* ATCC 27853 and *Escherichia coli* 25922).

The most common bacterial agents associated with wound infection include *Staphylococcus aureus*, coagulase-negative staphylococci, *Escherichia coli*, *Pseudomonas aeruginosa, Klebsiella pneumoniae*, *Streptococcus pyogenes*, *Proteus* sp., *Streptococcus* sp., and *Enterococcus* sp. [32].

The preliminary determination of antibacterial activity was evaluated qualitatively via the diffusimetric method (Kirby–Bauer) in the spot against the bacteria mentioned above. The bacterial strains were grown overnight on a PCA medium (Plate Count Agar, Oxoid, UK). Bacterial suspensions with a density of 0.5 McFarland (1.5 × 10^8^ colony forming units CFU/mL) were performed. The petri dishes prepared with 20 mL of PCA were inoculated with a bacterial suspension. The gel samples were exposed for 15 min to UV for sterilization. Subsequently, the 100 mg of the gel from samples were spotted on the inoculated petri dishes. The prepared plates were incubated for 24 h at 37 °C. After the incubation period, the antimicrobial activity of the samples should have been highlighted by the presence on the culture medium of a clear area (inhibition area), which would have been due to the inhibitory effect of the gels.

The results obtained by this method were not significant, because the presence of an inhibition zone on the culture medium was not observed. Since the qualitative method did not provide suggestive information for any antimicrobial activity of the amorphous hydrogel based on the AgNPs@CIN colloidal solution and vitamin C, the direct contact method was subsequently used. The quantitative method was not applied for the variants of Alg, Alg:COL, and Alg:COL:Gli because the qualitative method clearly confirmed the absence of an antibacterial effect (Figure 3) by the presence of bacterial growth.

Therefore, quantitatively antibacterial activity tested against *S. aureus*, *E. coli* and *Ps. aeruginosa* has been applied for basic hydrogel (Alg:COL:Gli:AV) and for hydrogels with AgNPs@CIN and vitamin C. The results were quantified by determining CFU/mL (Table 1).

Following the determination of the inhibitory effect of the amorphous hydrogels by the direct contact method, it was observed that the hydrogel variant Alg:COL:Gli:AV considered BASE does not show antibacterial properties. Moreover, the biocomposite facilitates bacterial multiplication. The behaviors of all the bacteria tested in the presence of this gel are similar; the CFU/mL is higher compared to the culture control, being higher by 1 log and in the case of *Ps. aeruginosa* by 2 log. Given that the experiment was performed in the absence of nutrients, it can be said that this variant promoted the bacterial growth.

In the case of the samples with silver nanoparticles, only the hydrogel Alg:Gli:COL:AV:AgNPs@CIN (2) showed significant antibacterial activity: the bacterial growth reduction was 2 Log (about 45%). In addition, a significant reduction in bacterial growth was observed in all bacterial strains treated with the hydrogel based on vitamin C, where the bacterial growth in the case of *S. aureus* was reduced by up to 4 Log (about 75% bacterial reduction) compared to the culture control. In the presence of the gel with 100 mg vitamin C content, the growth of the tested Gram-negative bacteria was 80–90% inhibited.

A bactericidal effect was observed for the hydrogel with a 200 mg vitamin C content against the Gram-negative bacterial strain tested (Figure 4).

The good antibacterial activity of the hydrogels with vitamin C it may be due to the much more acidic pH (4–4.5). This acidic nature could induce changes in the active sites of the enzymes causing changes to the peptidoglycan molecules in the structure of the bacterial cell wall [33]. A Gram-positive bacterium’s cell wall contains 3 to 20 times more peptidoglycan than that of a Gram-negative bacterium [34], which may explain why the tested Gram-negative bacteria are more susceptible to gel action.

Hence, the amorphous hydrogels with antibacterial activity against *S. aureus* ATCC 6538, *Ps. aeruginosa* ATCC 27853, and *E. coli* 25922 bacterial strains were hydrogels based on vitamin C and gels with AgNPs@CIN (95 µg Ag/formulation).

### 2.3. Proliferative Capacity

In accordance with the cytotoxicity scale, as per SR EN ISO 10993-5: 2009, a cell viability of over 80% corresponds to a non-cytotoxic character [35]. The quantification via the MTS method of the viability of the Hs27 fibroblast cells treated with the experimental samples revealed a non-cytotoxic behavior of the gels, showing a cell viability of over 80% (Figure 5). The in vitro evaluation of the cell viability by the MTS method also showed that the 2% sodium alginate solution induces a viability of 122%. The increased cell viability compared to the control suggests a good proliferation of the Hs27 cells. The addition of the fish collagen solution (0.01%) in the proportion described in the formulation did not induce a higher proliferation of Hs27 cells. The viability of the cells treated with Alg:COL was 108%. Of note, the small amounts of the fish collagen in the present formulation did not contribute significantly to the promotion of proliferation.

The addition of glycerol in the formula increased proliferation to 123%. Subsequently, the addition of 1% AV solution into the mixture of Alg:COL:Gli showed a cell viability similar to control. At this point, the BASE formulation does not affect cell viability, but does not promote the Hs27 cell proliferation either.

Between the gel variants with silver nanoparticles, the variant Alg:Gli:COL:AV:AgNPs@CIN (1) with low concentrations of nanoparticles (8 µg Ag/35 g gel) significantly promoted the proliferation of Hs27 cells, and with a cell viability of 130%. The pro-proliferative effect can be attributed to active compounds of the aqueous *Cinnamomum verum* extract, which covers the synthesized silver nanoparticles. In addition, a similar cellular proliferation was also observed in the case of the gel with the 100 mg vitamin C/formulation.

The amorphous hydrogel enriched with 200 mg vitamin C/formulation (0.57% vitamin C/formulation) showed a cell viability of 165%. This result is in agreement with the literature data, showing the beneficial impact of vitamin C on would healing. Lee et al. (2012) in vivo examined the effects of a vitamin C and Pluronic F127 (1 mg vitamin C/mL Pluronic F127 solution) composite on diabetic wound healing. The study showed that vitamin C incorporated into Pluronic F127 exhibits antioxidative and anti-apoptotic activities, which enhance epidermal and dermal maturation and collagen synthesis in diabetic skin [36]. Vivcharenko and Przekora (2021) summarized the beneficial effect of vitamin C incorporated into various polymeric composites intended for wound healing [37], such as the addition of vitamin C to a chitosan/agarose foam-like dressing [38], to a chitosan-based membrane [39], the addition of vitamin C and of propolis into a cellulose/PVA film [40], and a PDGF-BB biomaterial incorporated with L-ascorbic acid [38].

In conclusion, the gel Alg:Gli:COL:AV:AgNPs@CIN (1) and the gel variants with vitamin C promoted the proliferation of Hs27 cells, and makes them promising tools in wound healing.

### 2.4. Rheological Analysis of Amorphous Hydrogels

Since the Alg:COL:Gli:AV:AgNPs@CIN (1) hydrogel from the gel variants based on AgNPs@CIN and the Alg:Gli:COL:AV:Vit C (200) hydrogel from the vitamin C-based gel group showed the most pronounced cell proliferation, the present study aimed to evaluate the rheological behaviors of these amorphous hydrogels and compare them to the Alg, Alg:COL, Alg:COL:Gli, and Alg:COL:Gli:AV variants.

The linear viscoelastic region of the studied samples was determined using stress sweep tests at a constant frequency of 1 Hz and 37 °C, and the results are presented in Figure 6. In the case of Alg (2%), Alg:COL, and Alg:COL:Gli, the storage modulus (G′) was smaller than the loss modulus (G′′), indicating a liquid-like behavior. Meanwhile, the samples with Aloe vera (Alg:Gli:COL:AV, Alg:Gli:COL:AV:AgNPs@CIN (1) and Alg:Gli:COL:AV:Vit C (200) presented a solid-like character (G′ > G′′). At shear stresses above 5 Pa, all the studied samples showed the beginning of plastic deformation, marked by a decrease in the storage modulus.

The rheological fingerprint of the mixtures was studied using frequency sweep tests in the range from 0.1 to 16 Hz at a constant shear stress, and the results are summarized in Figure 7.

The rheological spectra indicated that the mixtures (Figure 6A–C) showed a liquid-like behavior (G′ < G′′) followed by a solid behavior at higher frequencies (G′ > G′′). Furthermore, the addition of collagen and glycerin in the simple alginate solution decreases the two moduli by an order of magnitude. In contrast, the presence of Aloe vera in the alginate mixtures causes the samples to exhibit a gel-like behavior (G′ > G′′) and leads to a significant increase in the values of the storage and loss moduli, making the gel more resistant to deformation (see Figure 6D–F). This behavior was observed over the entire frequency range. Furthermore, both the rheological moduli increase with an increasing frequency; the increases in G′ and G′′ are more evident at a high frequency (more than 10 Hz). This rheological response may be due to the fact that the hydrogel does not have time to reach the new equilibrium state, specific characteristic of soft gels 

In conclusion, the Alg:Gli:COL:AV, Alg:Gli:COL:AV:AgNPs@CIN (1), and Alg:Gli:COL:AV:Vit C (200) hydrogels have a viscoelastic gel texture with improved deformation resistance, mostly due to the presence of the Aloe vera solution.

## 3. Conclusions

Herein, we proposed a base formulation of amorphous hydrogels with no cytotoxicity, which can be enriched with additional compounds, such as AgNPs or vitamin C, for further biological effects. We showed that the addition of aloe vera into the gel is essential for the gel’s texture, as proven by the rheological tests. Although fish collagen type I was added to the base formulation to enhance cellular viability, the concentration used was too low to induce proliferation. The addition of AgNPs@CIN (8 µg Ag/formulation) at the BASE promoted Hs27 proliferation, but the developed hydrogel did not show antibacterial activity. Increasing the concentration of AgNPs@CIN (95 µg Ag/formulation) induced a significant antibacterial effect in the gel (the bacterial growth reduction was about 45%) but did not induce any cell proliferation. The vitamin C-based hydrogels manifested a strong antibacterial activity, as the variant with 200 mg of vitamin C showed a bactericidal effect against Gram-negative bacteria. In addition, this hydrogel variant remarkably potentiated the cells’ proliferation, with a cell viability of 165%. Hence, the experimental results offer a starting point for further in vivo studies of hydrogel formulae with both antibacterial and dermal regenerative properties.

## 4. Materials and Methods

The following materials were used in this study: sodium alginate (VWR Chemicals, Leuven, Belgium); AgNO_3_ (Sigma Aldrich, Saint Louis, MO, USA); NaOH (EMPLURA^®^, Darmstadt, Germany); Type I collagen, extracted in-house from tails of *Hypophthalmichthys molitrix* (silver carp) with a purity > 95% (method previously published [28]); Aloe vera powder (MAYAM, Berlin, Germany); Vitamin C (MAYAM, Germany); *Cinnamomum verum* bark (SONNENTOR, Salzburg, Austria); Silver nanoparticles synthesized by the green method with aqueous *Cinnamomum verum* extract (AgNPs@CIN); Glycerol/glycerin (Scharlau, Barcelona, Spain); purchased culture media specific to biological tests; human skin fibroblast cell line (HS27; CRL-1634-ATCC); and purchased ATCC bacterial strains and isolated strains from the collection of the Department of Microbiology of the Faculty of Biology (University of Bucharest, Bucharest, Romania).

### 4.1. Synthesis of Silver Nanoparticles by the Green Method

In the present study, we adopted the simple method of obtaining AgNPs by green synthesis with aqueous Cinnamomum verum extract. The aqueous cinnamon extract was obtained from ground bark, where 2.5 g of cinnamon was added into 100 mL of deionized water, followed by boiling for 5 min. After cooling, the extract was filtered and centrifuged (9000 rpm/15 min) and was then re-filtered through filter paper (retention range 5–13 µm). A 10 mM AgNO3 solution was used for synthesis of silver nanoparticles (as a metal precursor) and aqueous cinnamon extract (as a reducer) in a ratio of 1:10 (volume:volume). The mixture was magnetically stirred at 200 rpm/15 min, at a temperature range of 45–50 °C. The pH of the extract was adjusted from pH 5 to pH 12 using 1 M NaOH solution. 

### 4.2. Hydrogel Formulation

Experimental polymeric composites were mixtures based on NaAlg, type I collagen (isolated from silver carp tails) (COL), aloe vera powder (AV), glycerol (Gli), AgNPs@CIN and vitamin C. 2% sodium alginate (Alg), fish collagen (0.01%), and AV solution (1%) and were used to obtain the experimental variants. The basic mixture of amorphous hydrogels was Alg:Gli:COL:AV (BASE). The mixture was created at room temperature (24 °C) and the weight ratio of the solutions was 6:1 (Alg:Gli), while that of the volumes of the solutions was 2.4:1 (COL:AV). Homogenization of the compounds and the formation of the gel structure were performed by vortexing at 2500 rpm/5 min. Then, different variants of the hydrogel were obtained by adding the AgNPs@CIN colloidal solution (8 µg Ag/35 g gel and 95 µg Ag/35 g gel) and vitamin C (100 mg/35 g gel and 200 mg/35 g gel) to the BASE. The experimental variants of biocomposites were coded as follows: (i) Alg:COL, (ii) Alg:Gli:COL, (iii) Alg:Gli:COL:AV, (iv) Alg:Gli:COL:AV:AgNPs@CIN (1), (v) Alg:Gli:COL:AV:AgNPs@CIN (2), (vi) Alg:Gli:COL:AV:Vit C (100), and (vii) Alg:Gli:COL:AV:Vit C (200).

The obtained formulation was sterilized by UV light exposure for 30 min and stored in a refrigerator (4–6 °C) until further use.

### 4.3. Physicochemical Characterization of Silver Nanoparticles and Amorphous Hydrogels

The confirmation of AgNPs’ synthesisby employing the green method to reduce silver nitrate with aqueous *Cinnamomum verum* extract was performed by UV–vis absorption spectroscopy. The spectra were recorded with the Hitachi U-0080D UV–vis spectrometer. Absorption was measured in the wavelength range 200–800 nm; deionized water was used as a blank.

The morphological analysis of the synthesized AgNPs was performed by using the field emission scanning electron microscope (FE-SEM) Nova NanoSEM 630 (FEI Company, Hillsboro, OR, USA), equipped with EDX detector (EDAX TEAM ™, Pleasanton, CA, USA). The nanoparticles were washed with deionized water three times by centrifugation at 9000 rpm/15 min. The AgNPs were then resuspended in deionized water and spotted on SiO_2_ support followed by drying for 2 h in an oven at 50 °C for SEM and XRD (X-ray diffraction analysis) examinations. 

The Zeta potential, polydispersity index (PI), and hydrodynamic diameter of the AgNPs@CIN composites were analyzed using DelsaNano equipment (Beckman Coulter, Brea, CA, USA), which uses photon correlation spectroscopy (PCS), also called DLS (Dynamic Light Scattering) or ELS (Electrophoretic Light Scattering).

The phase identification and the evaluation of the crystal parameters were realized using a 9 kW Rigaku SmartLab X-ray Diffraction System (Japan) with rotating anode that employs Cu Kα1 radiation (λ = 1.5406 Å) in parallel-beam mode. During the XRD measurements, the incidence angle was kept at 0.5°, while the detector moved from 20° to 95°. The diffraction peaks’ indexing was conducted using the ICDD (International Centre for Diffraction Data) database.

Rheological analysis of hydrogels was performed on a Kinexus Pro Rheometer (Malvern, UK). The rheometer was equipped with parallel plate geometries (20 mm) and a gap of 0.5 mm. The working temperature was 37 °C (Julabo CF41 cryo-compact circulator). The linear viscoelastic region (LVER) was determined using stress sweep measurements at 1 Hz. The viscoelastic character was evidenced by frequency sweep test in the range of 0.1–16 Hz at a constant shear stress from LVER. The results were represented logarithmically.

### 4.4. Antibacterial Activity of Hydrogel Formulations

The antibacterial effect of hydrogels was evaluated against, *Staphylococcus aureus* ATCC 6538, *Pseudomonas aeruginosa* ATCC 27853 and *Escherichia coli* 25922 reference bacterial strains. Bacterial suspensions of 0.5 McFarland were performed in sterile saline solution from bacterial strains developed overnight on solid PCA medium (Plate Count Agar, Oxoid, UK).

A total of 300 mg gel was added in sterile Eppendorf tubes, and the samples were sterilized by exposure to UV light for 30 min. A total of 100 µL of bacterial suspension (0.5 McFarland) was added to the gel. In addition, the same amount of bacterial suspension for each test strain was used as culture control. The samples and the culture control were incubated for 24 h at 37 °C. Subsequently, 700 µL of sterile saline solution was added to the samples and vortexed for 60 s, to serve as starting solution for successive serial microdilutions. A total of 10 μL was spotted on PCA for the determination of Colony-Forming Units/mL (CFU/mL), which was calculated with the following formula (1): CFU/mL = N × 1/D × 10^2^(1)
where N = average of the colonies counted; 1/D = dilution at which the colonies were counted; 10^2^ = volume correction factor

### 4.5. Evaluation of Proliferative Activity

Cell proliferation induced by obtained hydrogels was tested on human foreskin fibroblast Hs27 (CRL-1634-ATCC) by spectrophotometric method using MTS assay. The cell culture was grown in DMEM/F12, supplemented with 10% fetal bovine serum (Corning) and 1% antibiotic-antifungal (Sigma-Aldrich) in a humidified atmosphere with 5% CO_2_ at 37 °C. A total of 10,000 cells were seeded in 96-well plates in triplicate for 24 h, and then treated with 100 mg gel of the various formulations presented above for additional 24 h. Untreated cells were used as controls. Triplicated additional wells, with no cells, were incubated with 100 mg gel to serve as background readings. At the end of the incubation period, the cell medium was removed and 100 µL of fresh medium and 20 µL of MTS (CellTiter 96^®^ Aqueous One Solution Reagent, Promega, Madison, WI, USA) reagent were added to each well. The sample plates were incubated at 37 °C for 3 h in a humidified atmosphere and 5% CO_2_. Cell viability was assessed by spectrophotometry, reading the OD (optical density) of the stained solution at 490 nm with the microplate spectrophotometer Anthos Zenyth 3100 (Anthos Labtech Instruments, GMBH, Austria). Proliferation of treated cells was compared to control with the formula (2):Cell viability (%) = [(DO sample − DO background sample)/(DO average control − DO average background control)] × 100(2)

## Figures and Tables

**Figure 1 gels-08-00604-f001:**
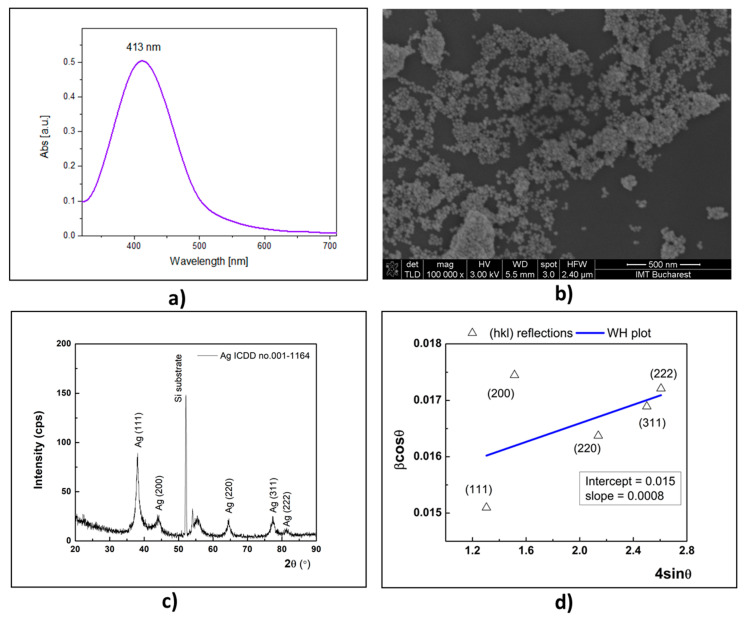
The characteristics of AgNPs@CIN: (**a**) UV–vis absorption spectrum of AgNPs@CIN; (**b**) SEM images of AgNPs@CIN; (**c**) XRD pattern of obtained AgNPs@CIN; (**d**) Williamson–Hall size-strain plot.

**Figure 2 gels-08-00604-f002:**
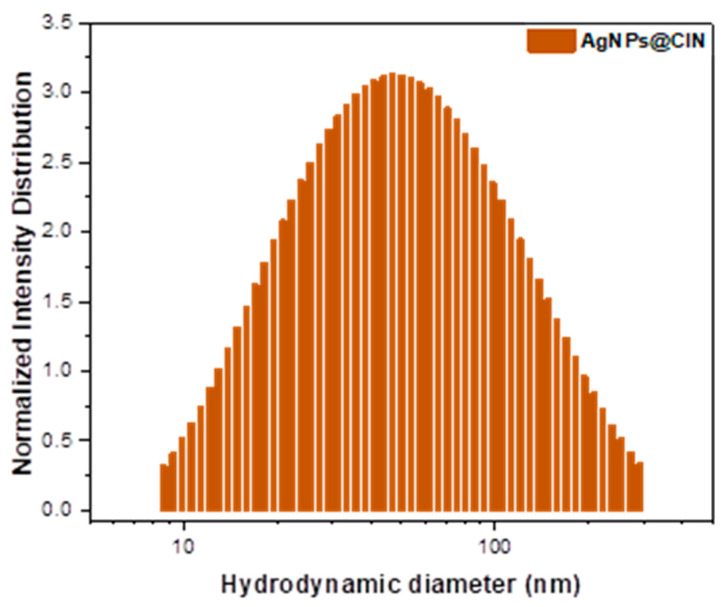
Distribution of hydrodynamic diameters of obtained colloidal silver nanoparticles (AgNPs@CIN).

**Figure 3 gels-08-00604-f003:**
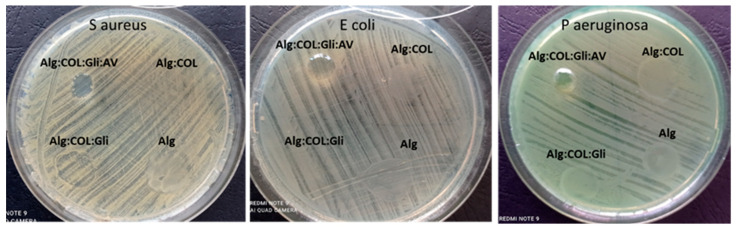
The antibacterial activity analyzed via qualitative method for Alg, Alg:COL, and Alg:COL:Gli against *S. aureus* ATCC 6538, *Ps. aeruginosa* ATCC 27853, and *E. coli* ATCC 25922 bacterial strains.

**Figure 4 gels-08-00604-f004:**
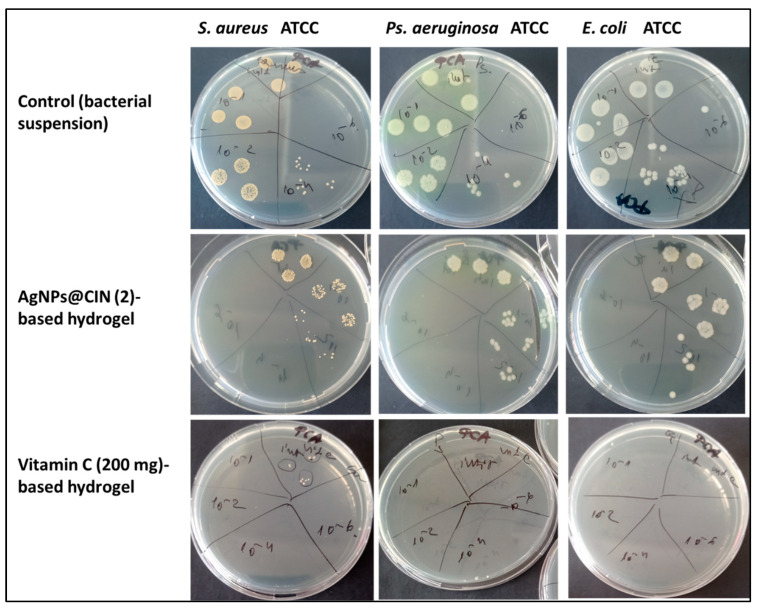
Antibacterial activity of the hydrogel based on AgNPs@CIN (95 µg Ag/35 g gel) and vitamin C (200 mg/35 g gel) against *S. aureus* ATCC 6538, *Ps. aeruginosa* ATCC 27853, and *E. coli* ATCC 25922.

**Figure 5 gels-08-00604-f005:**
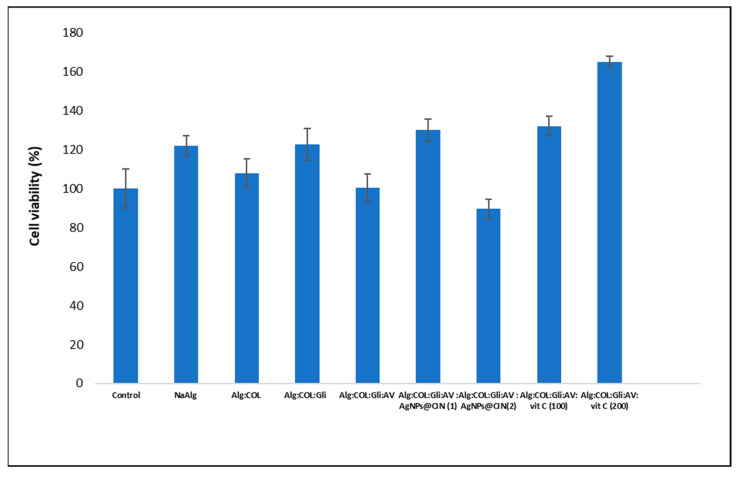
Cell viability of Hs27 cells treated with 100 mg/mL gel, determined spectrophotometrically by the MTS method. Error bars represent standard deviation (*n* = 3).

**Figure 6 gels-08-00604-f006:**
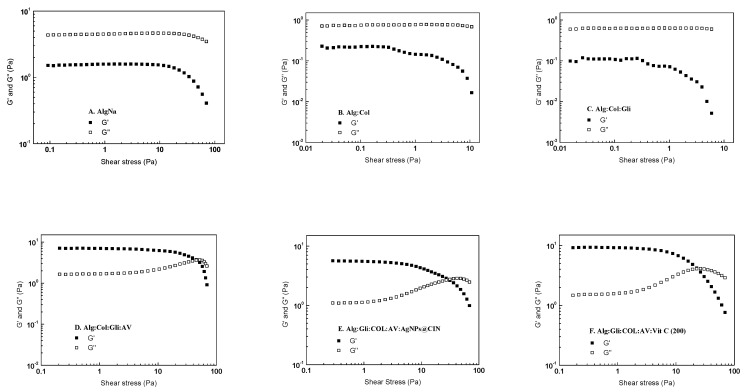
Dependence of G′ and G′′ moduli on the amplitude stress at 37 °C for: (**A**) Alg; (**B**) Alg:COL; (**C**) Alg:Gli:COL; (**D**) Alg:Gli:COL:AV; (**E**) Alg:Gli:COL:AV:AgNPs@CIN (1); (**F**) Alg:Gli:COL:AV:Vit C (200).

**Figure 7 gels-08-00604-f007:**
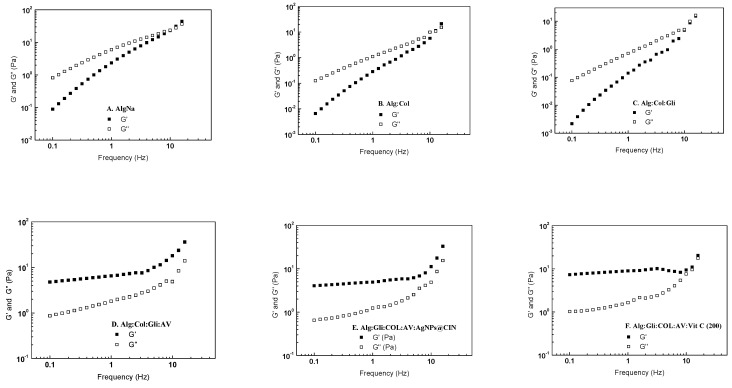
Dependence of G′ and G′′ moduli on the frequency at 37 °C for: (**A**) Alg; (**B**) Alg:COL; (**C**) Alg:Gli:COL; (**D**) Alg:Gli:COL:AV; (**E**) Alg:Gli:COL:AV:AgNPs@CIN (1); (**F**) Alg:Gli:COL:AV:Vit C (200).

**Table 1 gels-08-00604-t001:** Antibacterial activity of the obtained amorphous hydrogels.

Bacterial Strains	Log (CFU/mL)					
	Control (Bacterial Suspension)	Alg:Gli: COL:AV	Alg:Gli: COL:AV: AgNPs@CIN (1)	Alg:Gli: COL:AV: AgNPs@CIN (2)	Alg:Gli: COL:AV: Vit C (100)	Alg:Gli: COL:AV: Vit C (200)
***E. coli* ATCC 25922**	6.9	7.2	6.4	4.3	1	0
** *Ps. aeruginosa* ** **ATCC 27853**	6.6	8.5	6.3	4.2	2.2	0
** *S. aureus* ** **ATCC 6538**	6.7	7	6.5	4.8	3	2.4

## Data Availability

The data that support the findings of this study are available from the corresponding author upon reasonable request.

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
