# Peer review of "New Amorphous Hydrogels with Proliferative Properties as Potential Tools in Wound Healing"

_gels, 2022, doi:10.3390/gels8100604_

Round 1
Reviewer 1 Report
1. In the Abstract, Cinnamomum verum must be in italics; please check the whole manuscript that genus species must be in italics form.
Introduction lines no 58 to 63 must be rewritten; physical form limits 61 their application in certain clinical phases of lesions, so they have been replaced with advanced products that include wound dressings and permanent skin replacements / substatutes cant understand clearly.
Please check these lines, and put italics for the same: To our knowledge, no 140 data has been reported to show the proliferative or antibacterial activity of amorphous hydrogel based on AgNPs obtained by green synthesis with aqueous Cinnamomum verum extract. Moreover, amorphous hydrogels developed on variants with AgNPs@CIN and vitamin C haven’t been reported so far.
i requested the authors to go for grammar check to improve the language and to understand the readers
Author Response
Dear Reviewer 1 ,
Thank you for taking the time to review our article, for your opinion and comments.
We modified the English grammar in the text, and used the italic form for genus species.
All modifications are made with track changes, see the manuscript document.
Reviewer 2 Report
-
Author Response
Dear Reviewer 2,
Thank you for taking the time to review our article, and for your opinion.
We modified the English in the body of the manuscript, and this are performed with track changes.
Reviewer 3 Report
Authors have 21 developed a new formulation of amorphous hydrogel based on sodium alginate (NaAlg), type I 22 collagen isolated by us from silver carp tails (COL), glycerol (Gli), Aloe vera gel powder (AV), silver 23 nanoparticles obtained by green synthesis with aqueous Cinnamomum verum extract 24 (AgNPs@CIN) and vitamin C, respectively. The gel texture of the amorphous hydrogels was 25 achieved by the addition of Aloe vera, demonstrated by rheological analysis. Evaluation of cytotox-26 icity and cell proliferation capacity of experimental amorphous hydrogels were performed against 27 human foreskin fibroblast Hs27 cells (CRL-1634-ATCC). However, there are certain issues to be fixed prior to consideration for publication step.
1. Qualitative data is missing for;
i. DLS analysis
ii. antibacterial analysis of Ag containing gels
iii.Cell viability, cell images on the gels, or proliferation by live//dead assay
2. How did authors qualitatively access the crystal properties of silver and where is the Williamson-Hall size-strain plot?
3. Materials and methods need to be re-written to clearly mention the synthesized and purchased chemicals. Plus for synthesized materials, please provide some characterization for the successful synthesis.
4. Authors prepared different variants of the hydrogel by adding the AgNPs@CIN colloidal solution (8 μg Ag / 35 g gel) and also vitamin C (200 mg/35 g gel) to the BASE, how do they come up with 8 μg Ag / 35 g gel and 200 mg/35 g gel concentrations of AgNPs and vitamin C respectively? What about higher or lower concentrations than these?
5. There are a lot of grammar and typo errors ( Line 42-45, Line 160, Line 174, Line 196, Line 201, Line 241, Line 363, Line 365, Line 394, plus others) to be carefully eradicated.
Author Response
The answer is attached

Round 2
Reviewer 1 Report
The manuscript entitled "New amorphous hydrogels with proliferative properties as potential tools in wound healing' were revised and fulfilled all the requirements, so I requested the editor to accept the manuscript. Thank you
Reviewer 3 Report
The authors have addressed all the issues, manuscript can be accepted in its present form.